# Atomic-scale observation of nucleation- and growth-controlled deformation twinning in body-centered cubic nanocrystals

Li Zhong[1,2], Yin Zhang [3], Xiang Wang [1], Ting Zhu [3] ✉ & Scott X. Mao [1] ✉

Twinning is an essential mode of plastic deformation for achieving superior strength and ductility in metallic nanostructures. It has been generally believed that twinning-induced plasticity in body-centered cubic (BCC) metals is controlled by twin nucleation, but facilitated by rapid twin growth once the nucleation energy barrier is overcome. By performing in situ atomic-scale transmission electron microscopy straining experiments and atomistic simulations, we find that deformation twinning in BCC Ta nanocrystals larger than 15 nm in diameter proceeds by reluctant twin growth, resulting from slow advancement of twinning partials along the boundaries of finite-sized twin structures. In contrast, reluctant twin growth can be obviated by reducing the nanocrystal diameter to below 15 nm. As a result, the nucleated twin structure penetrates quickly through the cross section of nanocrystals, enabling fast twin growth via facile migration of twin boundaries leading to large uniform plastic deformation. The present work reveals a size-dependent transition in the nucleation- and growth-controlled twinning mechanism in BCC metals, and provides insights for exploiting twinning-induced plasticity and breaking strength-ductility limits in nanostructured BCC metals.

The mechanical properties of crystalline solids have been widely found to benefit from the formation of twins during plastic deformation[1-3]. However, plastic deformation in body-centered cubic (BCC) metals is usually governed by screw dislocations with high lattice resistance, as shown either in nanowires/nanopillars[4-11] with diameters as small as ~100 nm or bulk counterparts with grain/crystal sizes varying from dozens of nanometers to micrometers[12-17]. Deformation twinning in BCC metals is reported primarily under extreme deformation conditions such as low temperature and high strain rate[18-20]. BCC metallic nanostructures undergoing prevalent dislocation plasticity usually exhibit high yield strength but limited ductility due to their strong tendency to shear localization[12,17,21-23]. Recently, deformation twinning was observed to dominate the plastic deformation of sub-50-nm-diameter BCC nanocrystals under room temperature and low strain rate[24,25]. These studies suggest opportunities of harnessing the

twinning-induced plasticity in BCC nanostructures toward high mechanical performance, and thus underscore the importance of atomic-scale understanding on twin nucleation and growth, which have relied almost exclusively on computer simulations[26-33] and theoretical models[34,35] to date.

Deformation twinning in BCC metals is generally believed to be nucleation-controlled, as the energy barrier for twin nucleation is significantly higher compared to that for twin growth through migration of twin boundaries[28,36]. As a result, twin growth is expected to proceed rapidly once a twin embryo is formed[29,30,37], which effectively accommodates plastic strain and reduces shear stress, thus decreasing the probability of twinning events in its vicinity. However, incipient deformation twins in BCC metals were frequently found in the form of a high density of parallel twin islands aligned within slip bands[38,39]. In addition, a recent experiment revealed the formation of nanoscale

[1]Department of Mechanical Engineering and Materials Science, University of Pittsburgh, Pittsburgh, PA, USA. [2]SEU-FEI Nano-Pico Center, Key Laboratory of MEMS of Ministry of Education, Southeast University, Nanjing, China. [3]Woodruff School of Mechanical Engineering, Georgia Institute of Technology, Atlanta, GA, USA. ✉e-mail: ting.zhu@me.gatech.edu; sxm2@pitt.edu

twins during deformation of coarse-grained BCC metals under room temperature and low strain rate[40]. This result indicates that the limited twinning deformation in BCC metals could be attributed to not only the high critical stress required for twin nucleation, but also the even higher stress needed for twin growth (i.e., thickening and lateral expansion of a twin embryo). These observations call into question the conventional wisdom of nucleation-controlled deformation twinning in BCC metals.

In this work, we perform in situ atomic-scale TEM observations of plastic deformation in nanoscale single-crystalline BCC Ta, and find distinctive modes of *reluctant* versus *facile* twin growth depending on the diameter of nanocrystals. The sample elongation produced by twinning is in good agreement with the theoretical value of 41.4%. The size-dependent twinning modes result in different mechanical behavior of Ta nanocrystals, and are closely related to the twin boundary structures as well as to the competition between twin growth and dislocation plasticity. These findings provide insights into the twin growth mechanisms in BCC metals, and highlight the potential of exploiting deformation twinning to achieve high strength and ductility in BCC nanostructures.

## Results

### Growth-controlled twinning in a 23-nm-diameter Ta nanocrystal

Figure 1 and Supplementary Movie 1 show the twin nucleation and growth processes in a 23-nm-diameter single-crystal Ta nanobridge (Fig. 1a and its inset) under [001] tension, viewed along [1$\bar{1}$0]. Plastic yielding occurred at a tensile strain of ~3.2% by nucleation of a twin embryo from the left surface of the nanocrystal (Fig. 1b). The corresponding yield stress (i.e., twin nucleation stress) is ~4.6 GPa, as estimated with the elastic constant[41]. A close-up view reveals that the slender twin embryo (inset in Fig. 1b) has a total length of 8 nm and a sharp twin tip with a local thickness of 0.8 nm. At the twin tip, the upper (112) coherent twin boundary (CTB) consists of two bunches of atomic steps. Each bunch has a thickness of ~0.4 nm and involves three successive atomic steps, which correspond to three (112) twinning partials. The preferred bunching of three twinning partials on the CTB is also observed in our atomistic simulations (Supplementary Fig. 1). Such step bunching at the CTB was observed during subsequent twin growth (Fig. 1c–e). It is generally recognized that the crystallographic slip by a 1/2 <111> screw dislocation on a {110} plane is equivalent to that by three 1/6 <111> twinning partials on successive {112} planes. Hence, the above observation provides evidence in favor of the view that twin nucleation and growth are mediated by screw dislocations in BCC metals[35,38].

Contrary to the common view of rapid twin growth after nucleation[28–30,37], the twin embryo was observed to grow slowly by thickening and lateral expansion under increasing load (Fig. 1c–h and Supplementary Fig. 2). The twinned region (Fig. 1c–g) exhibits a Moiré pattern (indicated by a parallelogram of cyan dots in the inset of Fig. 1d), which is produced by interference between the overlapping matrix and twin lattices along the viewing direction (Supplementary Fig. 3a, b). The region with Moiré fringes appears with an approximately elliptical shape (outlined by a cyan dotted ellipse in Fig. 1c–g). This shape matches the projection of an inclined twin boundary inside the nanocrystal, given a nearly circular cross section of nanocrystals resulting from our synthesis approach[24]. The above analysis of twin shape is corroborated by a curved lower twin boundary in the TEM image (Fig. 1c–e; indicated by yellow dotted curves), which results from the intersection between an inclined twin boundary and the side surface of the Ta nanocrystal. As the twin grew, another region with Moiré fringes appeared close to the upper-left twin boundary (outlined by a magenta dotted ellipse in Fig. 1f, g; Supplementary Fig. 3c, d). The twin growth process from in situ TEM observations and the corresponding evolution of Moiré fringes in Fig. 1 are illustrated in Supplementary Fig. 4. An explanation of twin growth accounting for such observations is given in Discussion section.

The twin growth process was accompanied by a significant amount of local lattice straining, as indicated by the lattice contrast formed due to lattice bending and distortion in the vicinity of the growing twin structure (marked by red arrow heads in Fig. 1c–g). As a result, a misalignment angle up to 12° (Fig. 1g) was produced between the matrix above and below the twin structure. Due to lattice bending and distortion, (002) lattice planes were not clear enough to measure the lattice strain and thus tensile stress during twin growth. However, by tracking the overall nanocrystal elongation, we found no appreciable strain burst after twin nucleation, in stark contrast to the occurrence of a major strain burst at the yield point in sub-15-nm-diameter Ta nanocrystals (see below). The plot of tensile strain against time (magenta curve) in Supplementary Fig. 5 confirms that no marked discontinuity in loading occurred upon twin nucleation. Instead, the tensile strain and accordingly flow stress increased continuously during twin growth, evading the common instability of nanocrystals at the yield point[29,37]. The growing twin structure finally penetrated through the cross section of the nanocrystal under increasing load, giving rise to an abrupt elongation (~4%) (Fig. 1h). As a result, the region with Moiré fringes shrunk (Fig. 1i) and a clear fast Fourier transform (FFT) pattern appeared, indicative of formation of a 1/6[11$\bar{1}$](112) twin (inset in Fig. 1h). Hence, Fig. 1 demonstrates a dynamic process of growth-controlled deformation twinning without discontinuous strain burst in a nanocrystal of BCC Ta.

### Facile twin boundary propagation in sub-15-nm-diameter Ta nanocrystals

Figure 2 and Supplementary Movies 2-3 show twin growth in two sub-15-nm-diameter Ta nanocrystals under [001] tension, which features the facile migration of twin boundaries. In Fig. 2a–d, deformation twinning occurred in a 10-nm-diameter Ta nanocrystal, viewed along [1$\bar{1}$0]. Plastic yielding occurred in this nanocrystal at a higher stress of ~6.8 GPa (as opposed to ~4.6 GPa in Fig. 1) by nucleation of a deformation twin in one of the two equivalent twinning systems 1/6[$\bar{1}$11](11$\bar{2}$) or 1/6[1$\bar{1}$1]($\bar{1}$12). Twin nucleation was accompanied by a major strain burst of ~9% (Fig. 2c, see the dark cyan curve in Supplementary Fig. 5b, Supplementary Dataset 1) and further followed by a uniform elongation up to ~40% (Fig. 2d). Since the twinning direction is either [$\bar{1}$11] or [1$\bar{1}$1], its in-plane projection in Fig. 2a–d is parallel to the loading direction of [001] (indicated by a blue arrow in Fig. 2c). As a result, the twinning process produced only elongation along [001] with no change in length along [110], i.e., giving no change of the projected diameter of this nanocrystal. Furthermore, since the twinning plane, either (11$\bar{2}$) or ($\bar{1}$12), is not edge-on from the [1$\bar{1}$0] viewing direction, an atomically-resolved image of a twin lamella bounded by two CTBs could not be obtained from the current viewing direction. However, as schematically shown in Fig. 2g, the twinning process resulted in a reorientation of the nanocrystal from [001] to [$\bar{1}$10] in the loading direction, as well as a simultaneous reorientation of [1$\bar{1}$0] to [001] in the viewing direction, as previously demonstrated by MD simulations[29,37]. Such lattice reorientations are clearly shown in Fig. 2d where twinning in the nanocrystal has completed (see the corresponding diffraction analysis in Fig. 2e, f). The 40% elongation in Fig. 2d is also in good agreement with the theoretical value of 41.4% produced by twinning-induced crystal reorientation (with the corresponding twinning shear strain of $\sqrt{2}/2 = 70.7\%$). The fact that the twin growth stress is much lower than the twin nucleation stress (Fig. 2h and Supplementary Dataset 1) in the 10-nm-diameter nanocrystal demonstrates a nucleation-controlled twinning process, which is fundamentally different from the reluctant twin growth observed in the 23-nm-diameter Ta nanocrystal (Fig. 1 and Supplementary Fig. 2).

In order to directly observe twin boundary migration, Fig. 2i–l shows the deformation twinning process in a different nanocrystal with a diameter of 12 nm under [001] tension. In this case, the viewing direction is [100], as opposed to [1$\bar{1}$0] in Fig. 2a–d; the two

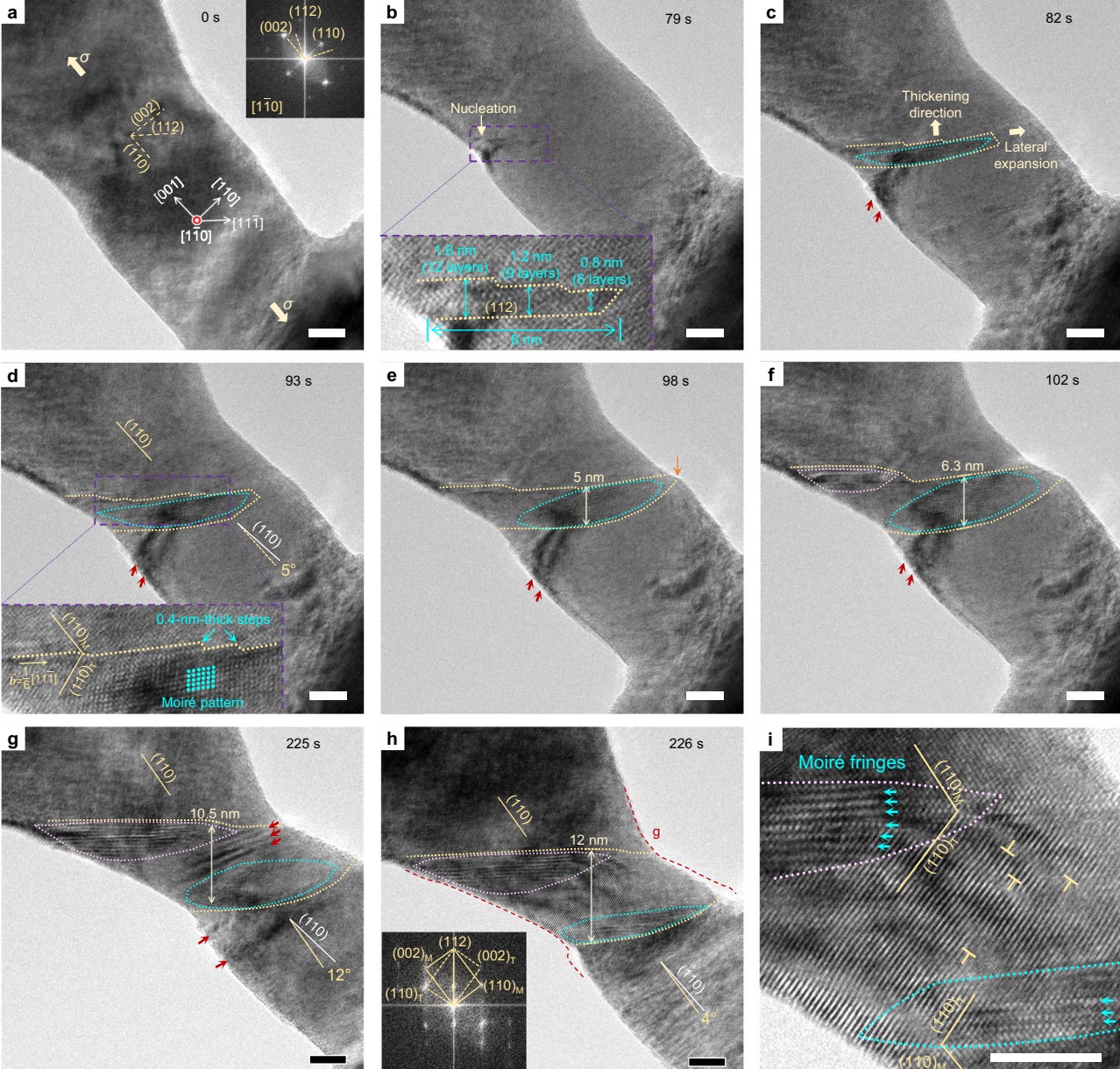

**Fig. 1 | Reluctant twin growth in a 23-nm-diameter Ta nanocrystal. a** An as-formed 23-nm-diameter Ta nanocrystal under [001] tension ($\sigma$) at a strain rate of approximately $10^{-3}\,s^{-1}$. The viewing direction is [1$\bar{1}$0] (inset in **a**). **b** Nucleation of a 1/6[11$\bar{1}$](112) deformation twin at the left-side surface of the Ta nanocrystal. A close-up view of the twin embryo (inset in **b**) shows a minimum thickness of six (112) layers at the twin tip and a width of 8 nm. The twin boundary consists of bunched steps; each bunch has three atomic-sized steps and a height of 0.4 nm. **c–h** Sequential images showing slow growth of a finite-sized deformation twin. The twin boundaries are tracked by yellow dotted lines. The twin tip reached the right-side surface of the nanocrystal (indicated by an orange arrow in **e**) when the twin has grown to a thickness of 5 nm. While the upper twin boundary is a CTB consisting of 0.4-nm-thick step bunches (inset in **d**), the lower twin boundary is inclined, which exhibits a curvature formed by intersecting the side surface of the Ta nanocrystal.

As a result, most regions of the twin lamella exhibited Moiré fringes (represented by cyan dots in the inset in **d**). The Moiré-fringed regions developed into shapes that match the projection of inclined interfaces (outlined by cyan and magenta dotted curves). Dark contrasts (indicated by red arrow heads) and a misorientation between the parent crystals sandwiching the twin indicate significant lattice distortion and bending near the twin, which was relieved after the twin finally penetrated through the diameter of the nanocrystal in **h**. The red dotted curves outline the nanocrystal shape one second earlier in **g** for comparison. The FFT (inset in **h**) demonstrates the diffraction pattern of a {112} twin. **i** High resolution TEM image of the deformation twin containing a high density of dislocations. Moiré fringes were found at both the top and bottom twin boundaries (indicated by cyan arrow heads). All scale bars are 5 nm.

viewing directions differ by a rotation of 45° around the tensile loading direction of [001]. In Fig. 2i, the two (1$\bar{1}$2) CTBs are close to an edge-on configuration. Movements of the two (1$\bar{1}$2) CTBs (Fig. 2j, m) were directly observed upon plastic yielding that featured the formation of a twin segment through the diameter of the nanocrystal with a strain burst to 15%. After this process, the subsequent

deformation involved facile migration of CTBs to an overall elongation of ~37% (Fig. 2k–l), similar to the twinning process in Fig. 2a–d. The drastic differences in the twin geometry and growth mode between Figs. 1 and 2 clearly indicate a size-dependent transition from the growth to nucleation-controlled twinning mechanism in BCC Ta.

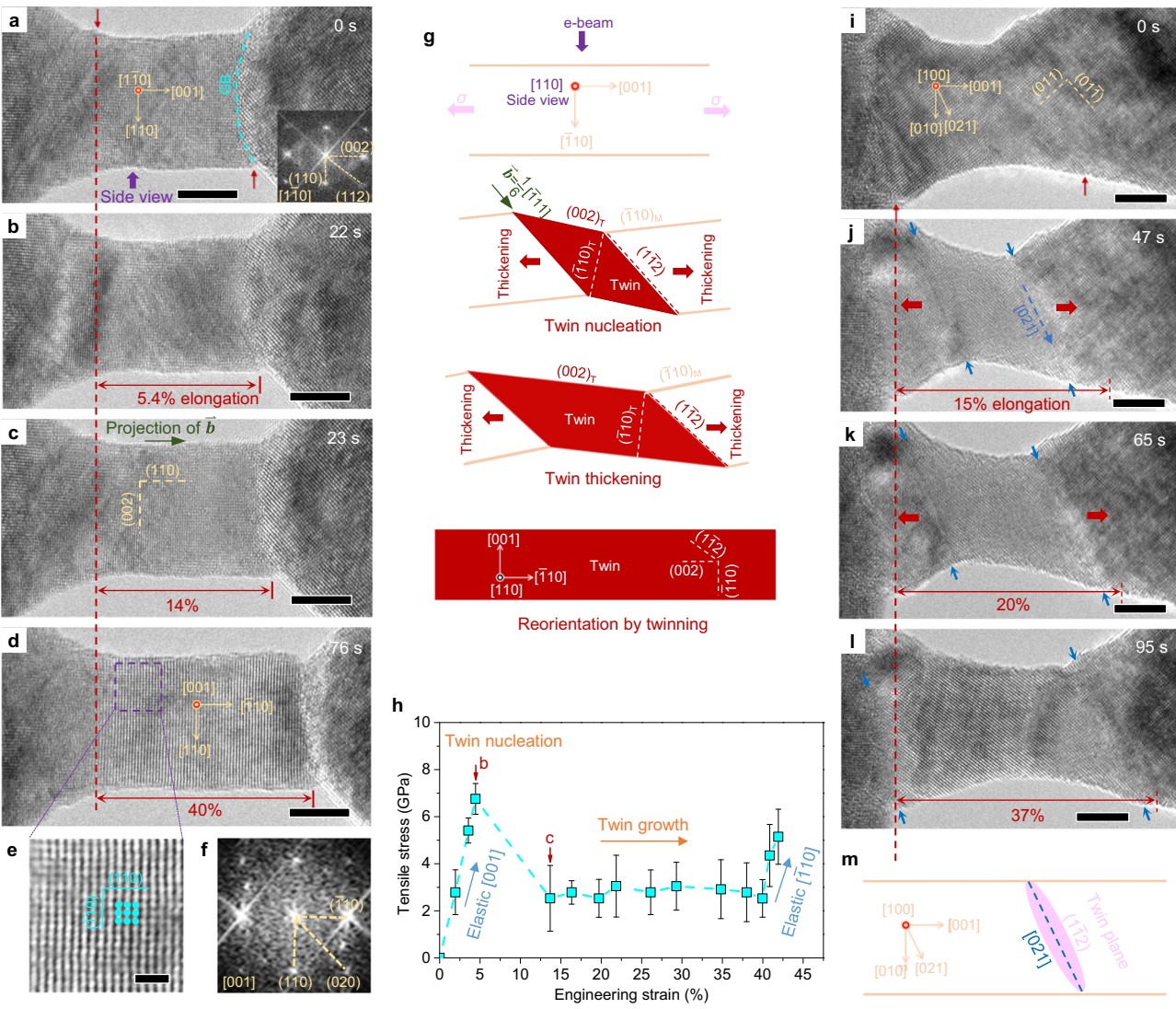

**Fig. 2 | Large ductility via facile twin thickening in sub-15-nm Ta nanocrystals.**
**a**, **b** A 10.4-nm-diameter Ta nanocrystal under [001] tensile loading at an average strain rate of ~5 × 10⁻³ s⁻¹. The viewing direction is [1̄10]. The nanocrystal was elastically stretched to a strain of ~4.7% before yielding **b**. The red arrows in **a** denote two reference points for tracking the elongation and the cyan dash line marks a grain boundary (GB) between the nanocrystal and the substrate. **c** Nucleation of a deformation twin with a twinning system of either 1/6[1̄11](11̄2) or 1/6[1̄1̄1](1̄1̄2), which is accompanied by a strain burst to 14%. The direction of the projected twinning Burgers vector is as indicated by a green arrow. **d** The nanocrystal finally reached an elongation of ~40%. **e**, **f** High resolution TEM image and FFT of the nanocrystal after deformation demonstrating twinning-induced reorientation.
**g** Schematic illustration of the deformation process shown in **a**–**d** with a viewing direction along the [110] zone axis (indicated by a purple arrow in **a**). One of the two possible equivalent twinning directions—1/6[1̄11](11̄2)—was selected.
**h** Corresponding stress–strain curve demonstrating a significantly lower stress for twin growth compared to those for nucleation of twin and dislocations. The stress is estimated based on the lattice strain along the loading direction measured from the high resolution TEM images and their FFTs, while the overall strain is measured from the elongation of the nanocrystal. Error bars represent standard deviations in the lattice strains measured from different locations of a nanocrystal. The red arrows mark the data points obtained from **b**, **c**. Source data are provided as a Source Data file. **i**–**l** Deformation twinning in a 12-nm-diameter Ta nanocrystal under [001] tensile loading with a viewing direction of [100]. The red arrows in **i** denote two reference points for strain measurement. Plastic deformation began with the nucleation of a deformation twin in **j**, followed by rapid migration of both twin boundaries tracked by two pairs of blue arrow heads in **k**. The twin boundaries are parallel to the [021] direction. The elongation reached ~37% as the two boundaries swept the whole nanocrystal in **l**. **m** Projection of the (11̄2) plane in a [001]-oriented nanocrystal viewed from [100] direction, which lies along the [021] direction, confirming that the twin boundaries observed in **j**–**k** are (11̄2) twin planes. The scale bars are 5 nm **a**–**d**, **i**–**l** and 1 nm **e**.

## Dislocation-mediated deformation when sheared in an anti-twinning sense

The asymmetry in twinning and anti-twinning is known to contribute significantly to the anisotropic mechanical properties of bulk BCC metals[10,30,42,43]. Here, we demonstrate that the asymmetry is an intrinsic property independent of size effect and thus plays an important role in the competition between dislocation plasticity and deformation twinning in BCC nanocrystals. While <001> tensile loading generates lattice shear along the normal "twinning" direction on the {112} twin plane (Figs. 1 and 2), <110> or <112> tensile loading produces lattice shear on the {112} plane with the highest Schmid factor in an "anti-twinning" sense. However, anti-twinning experiences higher resistance than twinning and thus tends to give way to dislocation plasticity[6,42,43] (Fig. 3 and Supplementary Fig. 6). As shown in Fig. 3 for a 17-nm-diameter Ta nanocrystal under [112] tensile loading, dislocation plasticity predominated. The events of dislocation slip were discrete, leading to a series of abrupt increases in the thickness of surface steps (e.g., two surface steps are tracked by red and blue arrows in Fig. 3b,

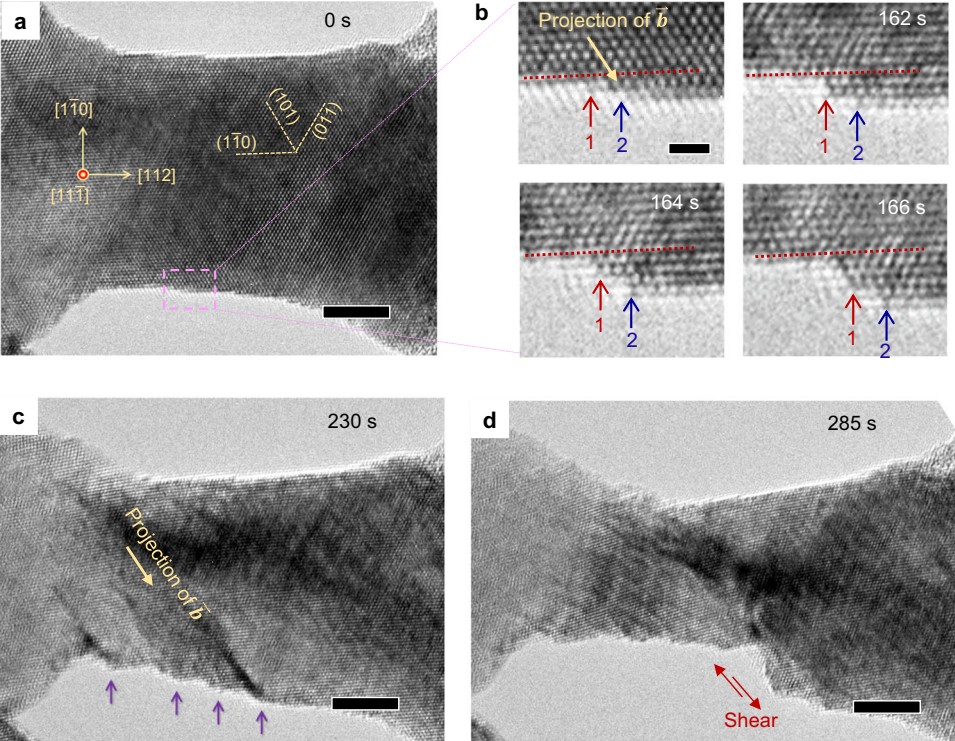

**Fig. 3 | Dislocation-mediated localized slip in Ta nanocrystals under <112> tensile loading. a** Tensile straining of a 17-nm-diameter Ta nanocrystal in the [112] direction at an average strain rate of ~$10^{-3}$ s$^{-1}$. **b** Sequential zoom-in images showing the thickness evolution of two surface steps (tracked by red and blue arrows) due to slip by nucleation and escape of 1/2[$\bar{1}$11](101) dislocations. **c**, **d** Localized slip steps led to the formation of a terraced surface profile (marked by purple arrows in **c**), which promoted the onset of shear localization in **d**. The scale bars are 5 nm **a**, **c**, **d** and 1 nm **b**.

respectively). Owing to the lack of surface diffusion to maintain a smooth surface contour, dislocation plasticity in nanoscale metals gives rise to localized slip[44], which leads to the formation of thick surface steps (indicated by purple arrows in Fig. 3c) and subsequent plastic instability due to shear localization (Fig. 3d). Compared to the terraced surfaces generated by dislocation-mediated slip (Fig. 3c), deformation twinning yields relatively smooth surfaces and is more resistant against shear localization. As dislocation plasticity prevails, increasing the aspect ratio of tensile samples further promotes localized slip and aggravates plastic instability[45], as observed in relatively long Ta nanocrystals under <112> and <110> tensile loading (Supplementary Fig. 6). Such "brittle" behavior largely limits the tensile ductility of BCC nanostructures and highlights a need to promote the twinning-induced plasticity for enhancing their ductility.

## Discussion

Understanding the atomistic mechanism of deformation twinning in BCC metals has largely relied on theoretical studies in the past[26,28,32,34,35]. Direct atomic-scale observation of deformation twinning was only made possible by recent advances in experimental techniques such as in situ TEM[24,40]. Here, our in situ observations reveal a dynamic process of growth-controlled deformation twinning that features the Moiré fringes resulting from inclined twin boundaries, which are associated with the formation of a finite-sized twin structure in relatively large Ta nanocrystals (Fig. 1). This twinning mode contrasts the nucleation-controlled deformation twinning in relatively small Ta nanocrystals, where a twin segment between two parallel CTBs penetrates through the diameter of Ta nanocrystals, producing homogeneous deformation across the twin plane (i.e., diameter) (Fig. 2). In the former case, a finite-sized twin structure generates highly strained regions in its vicinity (Fig. 1c–g; schematically shown in Supplementary Fig. 4d). The geometrical confinement from the matrix

in turn raises the driving stress needed for twin thickening (magenta curve in Supplementary Fig. 5b, Supplementary Dataset 1) to a level that is sufficient to activate dislocation plasticity, as evidenced by dislocations observed in the twinned lattice (Fig. 1i). In addition, due to the lack of a CTB at the lower inclined twin boundary, twin growth mainly proceeds one-sided at the upper CTB (Fig. 1b–g and Supplementary Figs. 4b–d). This is because thickening downwards requires nucleation of another CTB or migration of the inclined twin boundary. These processes are slow, but can gradually build up high lattice strains and thus strain energies to overcome the energy barrier of twin growth (Fig. 1g). Once the energy barrier is overcome, thickening downwards at the inclined twin boundary leads to traversing of the twin across the nanocrystal diameter for releasing geometrical confinements, and therefore proceeds rapidly (within 1 s in Fig. 1g, h).

To understand the reluctant growth of a finite-sized twin confined by the matrix, Fig. 4 and Supplementary Movies 4 and 5 show the results of molecular dynamics (MD) simulations of twin nucleation and growth mediated by cross-slip of screw dislocations in BCC Ta. While the suggestion of screw-mediated deformation twinning was found in the literature[32], achieving this through atomistic simulations has been challenging, due to a lack of in-depth understanding of the atomic-scale processes involved. Inspired by the present in situ TEM observations, we conducted MD simulations to realize the continued migration of a twin boundary mediated by screw dislocations. Figure 4a shows an array of 1/2 < 111 > {110} screw dislocations in a simulation cell of single-crystal Ta. Under an increasing anti-plane shear load, this array of screw dislocations glided to the left. However, the mobility of screw dislocations was not high enough to accommodate the increasing applied strain. Consequently, the stress increased, triggering the transformation of the leading screw dislocation into three 1/6 < 111 > {112} twinning partials on successive {112} layers, thus producing a twin embryo (Fig. 4a). Subsequent twin

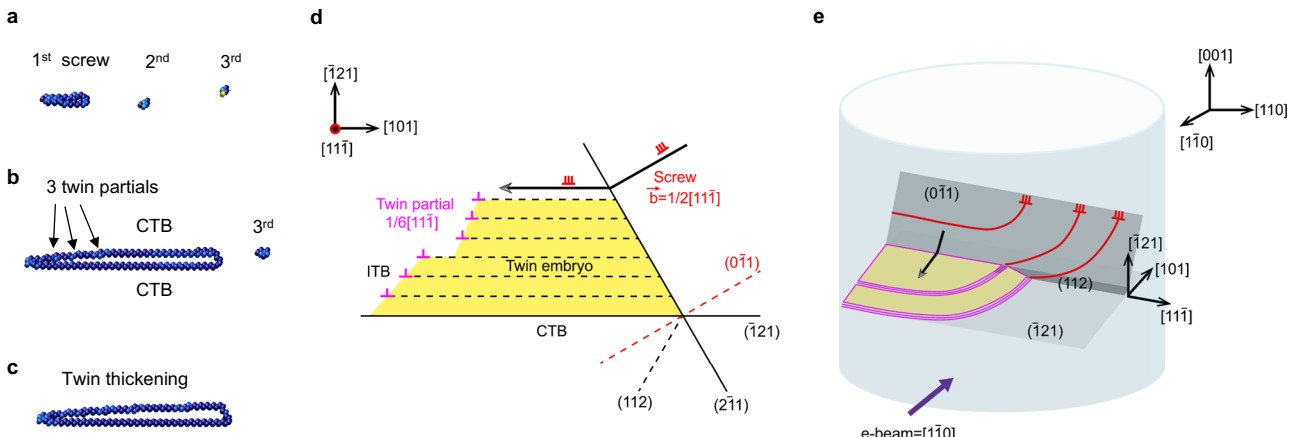

**Fig. 4 | Atomistic mechanism of nucleation and growth of deformation twins mediated by screw dislocations in BCC Ta. a** MD image showing an array of three gliding 1/2 < 111 > {110} screw dislocations under an applied anti-plane shear stress; nucleation of a twin embryo occurs through dissociation of a 1/2 < 111 > {110} screw dislocation into three 1/6 < 111 > {112} twinning partials on successive {112} layers. **b, c** MD images showing twin growth through cross-slip of 1/2 < 111 > {110} screw dislocations onto the CTB of the twin embryo. Each cross-slipped 1/2 < 111 > {110} screw dislocation becomes three 1/6 < 111 > {112} twinning partials at the CTB, as indicated in **b**. Atoms are colored by the central symmetry parameter showing the dislocation core and twin boundary. **d** Schematic of twin nucleation and growth through cross-slip of 1/2 < 111 > {110} screw dislocations; each cross-slipped screw dislocation (represented by three adjacent red symbols ⊥) is transformed into three twinning partials (represented by pink symbols ⊥) on successive {112} layers at the CTB, resulting in twin growth, The leading edge of the growing twin embryo consists of aligned twinning partials on successive {112} layers, forming an ITB. **e** Schematic of a corresponding three-dimensional process of nucleation and growth of a finite-sized twin structure inside a strained Ta nanocrystal. A unit process of twin growth still involves the cross-slip of a screw dislocation that dissociates into three twinning partials (pink lines) on the CTB. Several ITBs and CTBs form to enclose the finite-sized twin structure. Dislocation segments (red lines) that have not cross-slipped can continue to glide on their original slip planes near the finite-sized twin structure.

growth occurred through the cross-slip of follow-up 1/2 < 111 > {110} screw dislocations onto the CTBs of the twin embryo (Fig. 4b, c). Similar to twin nucleation, each cross-slipped screw dislocation became three twinning partials on successive {112} layers on a CTB, resulting in the thickening of the twin embryo, as schematically illustrated in Fig. 4d. The leading edge of this growing twin embryo consists of twinning partials aligned on successive planes, forming an incoherent twin boundary (ITB). Propagation of the ITB and accordingly lateral expansion of the twin embryo occurred through the movements of these twinning partials into the surrounding matrix, which required a continued increase of applied load, reflecting the reluctant twin growth confined by the matrix. The nucleation of screw dislocations was not simulated by MD, as they were considered to nucleate from nearby sources with relatively low energy barriers, such as on the free surface of Ta nanocrystals.

The above MD processes of dislocation-mediated twin growth are believed to underlie the in situ observations of reluctant twin growth in relatively large Ta nanocrystals. Since the applied load was not sufficiently high, the supply of twinning partials for twin growth relied on the continued nucleation and cross-slip of screw dislocations. Hence, the twin growth mediated by screw dislocations was reluctant, because the increasing load was necessary for the continued nucleation and cross-slip of these screw dislocations. To extend the above MD results obtained in a quasi-two-dimensional setup, Fig. 4e illustrates a similar three-dimensional process of nucleation and growth of a finite-sized twin structure inside a Ta nanocrystal. The key unit process still involves the cross-slip of a screw dislocation that dissociates into three twinning partials on the CTB. Several ITBs and CTBs form to enclose the finite-sized twin structure, and their movements also require an increasing load, thus reflecting the reluctant twin growth confined by the matrix. Meanwhile, dislocation segments that have not cross-slipped can continue to glide on their original slip planes near the growing twin. This response is consistent with the common observations of dislocations close to deformation twins[35].

As the diameter of Ta nanocrystals was reduced to sub-15 nm, the dislocation nucleation stress was likely elevated due to the well-recognized effect of "smaller being stronger". As a result, a high load could be attained to trigger the formation of a twin segment that penetrates through the diameter of small Ta nanocrystals. This twin segment facilitates the facile migration of CTBs without the need for screw dislocations. In these relatively small Ta nanocrystals (Fig. 5a), the formation of a twin segment that penetrates through the sample diameter may benefit from an increasing tendency for twinning partials to cross the nanocrystal as its size decreases. Hence, in sub-15-nm-diameter Ta nanocrystals, the high yield stress (Supplementary Fig. 7 and Supplementary Dataset 1) drives the nucleated twinning partials to quickly glide through the nanocrystal. Once the twinning partials approach the opposite surface, the image force[46] further facilitates lateral twin expansion, resulting in the formation of a twin segment bounded by two CTBs (Fig. 2j). Without the matrix confinement, the stress required for growth of the twin segment is much lower compared to those for the nucleation of twins and dislocations, as shown in Fig. 2h. As a result, twin thickening proceeds via rapid migration of both CTBs (Fig. 2j–l), in stark contrast to reluctant twin growth in relatively large Ta nanocrystals (Fig. 1b–g). This size-dependent competition between twin growth and dislocation plasticity is schematically shown in Fig. 5b.

To compare the twinning behavior in nanocrystals of different lattice structures, the present BCC Ta nanocrystals exhibit a size-dependent transition from the growth- to nucleation-controlled twinning mechanism. This is largely because the continued supply of twinning partials is difficult, and the movement of twinning partials on twin boundaries is slow for twin growth in large Ta nanocrystals. As a result, the applied load has to continuously increase to nucleate dislocations, drive cross-slip, and move twinning partials. For FCC crystals, such a size-dependent transition from growth- to nucleation-controlled deformation twinning is absent in various experimental studies[47–49], indicating that twin growth is less difficult once the applied load has reached the critical value for twin nucleation. On the other hand, for HCP crystals, a different size-dependent twinning behavior was observed[50]. As the size of an HCP Ti alloy is reduced to below one micrometer, dislocation slip replaces twinning to dominate plastic deformation. A 'stimulated slip' model was used to explain this size dependence of deformation twinning.

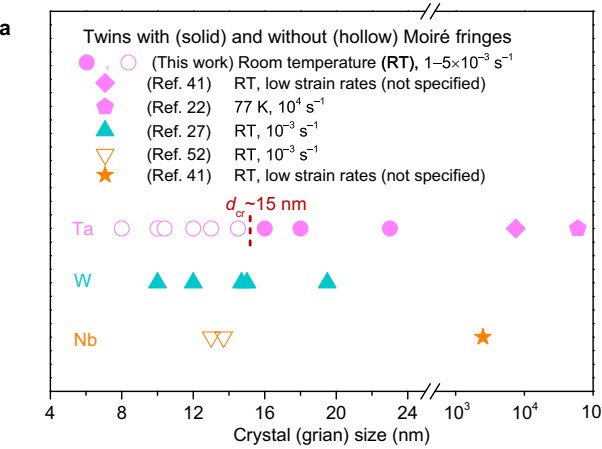

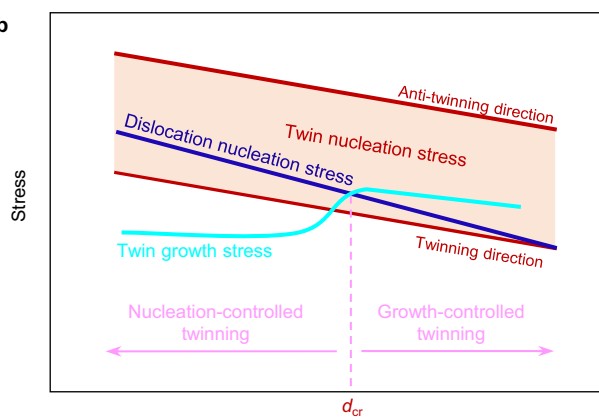

**Fig. 5 | Size-dependent deformation twinning in Ta nanocrystals. a** Summary of results from the present and past experiments showing a close relationship between twin boundary types and crystal/grain size. Solid and hollow data points, respectively, represent the existence and absence of Moiré fringes in the twinned region. No transition in twin boundary type is observed in W, indicating that the critical crystal size for W may be extremely small. **b** Size-dependent twin growth of BCC Ta in the size regime where plastic deformation is mainly mediated by twinning (very likely in the sub-100-nm regime based on previous results). The colored region represents the distribution of the twin nucleation stress along various loading directions, with the lower and upper bounds corresponding to the twin nucleation stresses under loading conditions that mostly favor slips in the twinning and anti-twinning directions, respectively. "$d_{cr}$" represents the critical diameter above which twin growth is frustrated by the increasing flow stress required for sustained twin propagation due to slow lateral twin expansion and formation of inclined twin boundaries. As a result, twin growth can be overwhelmed by dislocation plasticity above $d_{cr}$, which limits twinning-induced plasticity.

Moiré fringes associated with the twinned region have been frequently observed in BCC metals from sub-100-nm nanocrystals to coarse-grained bulk counterparts over a wide range of strain rates and temperatures[19,24,40,51] (Fig. 5a). These Moiré fringes suggest that the formation of inclined twin boundaries associated with finite-sized twin structures is a common phenomenon that limits the incipient twin growth in BCC metals. Inclined twin boundaries were observed in almost all samples tested[24] and the associated geometrical confinement contributes to slow twin thickening and a tendency for detwinning upon unloading[52] (see an example in a W nanocrystal in Supplementary Fig. 8). Several recent observations of nanoscale deformation twins with inclined twin boundaries in coarse-grained Ta and Nb under room temperature and low strain rates[40]—conditions that have been known to suppress deformation twinning—provide examples of correlating the limited deformation twinning processes with reluctant twin growth. Contrary to the dislocation starvation in nanocrystals, the growth-controlled twinning provides an opportunity to prevent the abrupt deformation instability after plastic yielding, rendering the small-sized materials to gain stable strain hardening. Note that the dependence of twin boundary type on size is also observed in other BCC metals such as Nb[51] (Fig. 5a), suggesting that the size-dependent deformation mechanism of Ta (Fig. 5b) may be generally applicable to nanoscale BCC metals. In addition, considering the similar roles played by small crystal size[47,48] and grain size[26,53] in facilitating deformation twinning, facile twin growth may also contribute to the ductility of bulk nanocrystalline BCC metals with grain sizes in the nucleation-controlled regime of deformation twinning (Fig. 5b). As a result, the present work reveals the critical role of crystal size in the deformation mechanisms and mechanical properties of BCC metals, and thus sheds light on strategies of exploiting twinning-induced plasticity to enhance the strength and ductility of nanostructured BCC metals.

## Methods
### In situ straining experiment
High purity Ta (99.98%) and W (99.98%) provided by ESPI Metals were used for mechanical testing in the present work. In situ straining tests were performed with a Nanofactory scanning tunneling microscope (STM) holder inside an FEI Tecnai F30 TEM. Two sharp Ta nano-tips at the fracture edge of Ta substrates were brought into contact and welded together by applying either an electric pulse[54] or a constant voltage[24], thus producing a Ta nanocrystal that bridges two Ta substrates for mechanical testing. The same approach was used to perform in situ straining tests on nanoscale W. Specimens prepared via the above approach are found to have a near-circular cross-section (Supplementary Fig. 9) with comparable geometries and surface defect characteristics (Supplementary Fig. 10). During in situ straining tests, the nanocrystals were viewed along low-index zone axes, such as <110> and <111>, to achieve atomically resolved observation on deformation twinning and dislocation processes. The strain rates were controlled by adjusting the retracting speed of the piezo-manipulator (i.e., piezo-head in Supplementary Fig. 5a) of the STM holder in the range of $0.001-0.005 \text{ s}^{-1}$.

### Atomistic simulation
We performed MD simulations of the nucleation and growth of deformation twins in single-crystal Ta using LAMMPS with an angular-dependent potential[55]. The simulation cell in Fig. 4 has the geometry of $37.6 \text{ nm} \times 33.1 \text{ nm} \times 3.5 \text{ nm}$. A periodic boundary condition was imposed along the $Z/[111]$ direction. Three screw dislocations of the $1/2 <111> \{110\}$ type were embedded in the central region of the Ta crystal by imposing their analytical displacement field. To apply an anti-plane shear load, we moved a boundary layer of atoms at the top surface of the simulation cell, while fixing a boundary layer of atoms at the bottom surface. The system's energy was minimized through the conjugated gradient method to mimic MD at low temperatures of -10 K. We obtained similar results of screw-mediated twin nucleation and growth using simulation cells with different initial dislocation positions. The central symmetry coloring scheme is used to visualize the dislocation core, CTB, and twinning partial, unless specified otherwise. In addition, dissociation of a screw dislocation into three twinning partials at the CTB was simulated in a Ta nanowire with a diameter of 16 nm and a length of 40 nm, as shown in Supplementary Fig. 1.

## Data availability
Data supporting the findings of this study are included within the article and Supplementary Information files. Source data are provided

with this paper. Additional data related to the current study are available from the corresponding authors upon request.

## Code availability

The code is available from the corresponding authors upon request.

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

## Acknowledgements

S.X.M. acknowledges support from NSF CMMI 1760916 through University of Pittsburgh. T.Z. acknowledges support from NSF CMMI 1762511 and DMR 2004412 .

## Author contributions

L.Z. and S.X.M. conceived and designed the experiments. L.Z. performed the in situ TEM experiments and wrote the manuscript. Y.Z. and T.Z. carried out computer simulations. X. W, T.Z. and S.X.M. revised the manuscript. All authors contributed to discussion of the results.

## Competing interests

The authors declare no competing interests.
