## [Peer Review File · Nature Communications]

Atomic-scale observation of nucleation- and growth-controlled deformation twinning in body-centered cubic nanocrystalsREVIEWER COMMENTS

Reviewer #1 (Remarks to the Author):

In the manuscript titled "Atomic-scale" observation of nucleation- and growth-controlled deformation twinning in BCC nanocrystals", the authors performed in situ atomic-scale TEM and atomistic simulations and reported the mechanism transition of deformation twinning from being "reluctant" or "facile" in BCC nanocrystals. In coarser BCC nanocrystals (>15 nm) deformation twinning is suppressed, whereas twinning is easier in finer BCC crystals (<15 nm). The new knowledge generated from this work focuses on the size-dependent twin nucleation and propagation mechanisms in BCC metals. The experiments are well designed. Results are beautiful. The paper is well written and I recommend acceptance after minor revision.

Here are some detailed comments for the authors to consider:

1. In the introduction part, maybe it is a good idea to mention the theoretical strain value deformation twinning in BCC metal could provide. The experimental results acquired later may be compared to the theoretical value. (I see it was mentioned in the results and discussion session. The authors may want to move the information up.)
2. In the first scenario (23 nm sample) at the twin embryo stage, is it possible to draw a Burgers circuit to unambiguously identify the Burgers vector of the twin dislocations?
3. It would be great if the authors can include a short paragraph in the manuscript discussing the transition of mechanisms here that may be applicable to twin boundaries in FCC and HCP metals. (The screw dislocation-mediated twin growth vs. facile growth the authors discussed here reminds me of the "shear" vs. "shuffle" debate in Mg twin growth. Maybe something in common may be drawn here?)
4. The authors selected Ta, which has a high brittle to ductile transition temperature compared to many other BCC metals. Do authors think other BCC metals but with lower brittle to ductile transition temperatures will exhibit similar observations and also manifest the mechanisms discovered here?

Reviewer #2 (Remarks to the Author):

For several reasons, the Introduction is too general or not specific enough. Mechanical twinning, TWIP effect, and these for different crystal structures (FCC, BCC, HCP) are put on the same level, without necessarily describing the differences. The authors state that the activation of mechanical twinning is in many cases beneficial. It is not clear either what the distinction is between macro- and nanostructures. First of all, in some materials, mechanical twinning is activated to compensate for the lack of gliding systems of dislocations. This is the case for HCP and some BCC. This is quite different from the TWIP effect active in FCC. This mix between classical and nano, twinning and TWIP, BCC and co, etc. does not reflect an adequate perspective of the role of mechanical twinning as a plastic deformation mode.

At line 58, the authors state "This result indicates that the limited TWIP effect in BCC metals may not only be attributed to the high critical stress required for twin nucleation, but also to the slow process of twin growth." This sentence is barely comprehensible. What do the authors mean by 'growth'? Is it lengthening or thickening? Considering for example the TWIP effect in some titanium alloys like Ti-15%Mo, this statement is difficult to consider.

The results presented are interesting, but I have some doubts about the consistency of the complete story and the accuracy of the measurements made in relation to the conclusions drawn. The procedure used for the fabrication of the samples used for the in-situ traction is quite particular, in relation to the nature of the material studied (Ta). But the consequence is a geometry of the tensile specimens that

does not correspond in all likelihood to a uniaxial tensile test and presents geometrical defects that undoubtedly influence the conditions of nucleation and growth of the twins. From what it can be seen on the different pictures illustrating the traction, the specimens are strongly constrained and it is quite sure that the stress state is not the same when the thickness changes. For example, Fig.1 and associated movie strongly looks like a localization mechanism due to a stress state resulting from some confinement. It is indeed quite evident that the width has an influence once considering the coordinated movement involved in growth as shown on Fig.2. As a consequence, the concluded crystal size effect could be a geometry effect.

Still considering the reported cases of twin nucleation, an accurate characterization of the configuration of the defect(s) bringing the embryo would be needed to understand the different reported situations. It would worth elucidating if these defects result from the 'machining' of the specimens or if they are intrinsic defects.

Line 118, Looking at Fig S5 and Fig1, they are not presenting the same thing, contrarily to what is suggested.

In the beginning of the Discussion, it is stated that mechanical twinning 'replaces' dislocation glide. This is thus not a TWIP effect, which involves a very large dislocation activity.

Reviewer #3 (Remarks to the Author):

Review report

Journal : Nature Communications

Title: Atomic-scale observation of nucleation- and growth-controlled deformation twinning in body-centered cubic nanocrystals

In BCC nanowires with size less than 50 nm, twinning is an important mode of plastic deformation. However, still there is a lack of understanding on the details of twin nucleation and growth, and size effects on twinning related phenomena, especially from the experimental perspective. In this regard, the present paper makes a significant improvement towards our understanding of twinning in BCC nanowires. Using in-situ TEM and atomistic simulations, the authors show a transition in twin growth mechanism with respect to nanocrystal size. They show that, in relatively smaller size nanocrystals (< 15 nm), the twinning is nucleation controlled and hence it's growth is facile, whereas in nanocrystal with size > 15 nm, the twinning is growth controlled (reluctant twin growth). Apart from this, the authors also show a dislocation controlled plasticity in nanocrystals having an antitwinning sense on the most favoured {112} planes. The authors clearly explained and discussed all the results in detail and provided the reasons for the observed transition from facile to reluctant twin growth with increasing size. Overall, an excellent paper and it can be accepted after incorporating the following minor suggestions/questions;

1. I feel that it would nice, if the authors show the transition quantitatively. The authors have shown that in nucleation-controlled twinning, twin growth stress is much lower than the twin nucleation stress (Figure 2h). Similarly, if possible, it should be shown that, in growth controlled twinning, the twin migration stress or stress during twin growth should be greater than the twin nucleation stress. If not experimentally, it is definitely possible through atomistic simulations. Previous study focussed on twinning and twin growth in BCC Fe nanowires (Comput. Mater. Sci. 104 (2015) 76-83) has shown that upto 24.27 nm size, the Fe nanowires show nucleation controlled twinning. As a result the migration stress is always lower than the twin nucleation stress up to the size of 24.27 nm. Further, limited ductility has been reported in relatively thicker nanowires due to difficulty in twin growth along

the nanowire length.

2. In Figure 1 (c-f), the authors have highlighted some distortions in the vicinity of twin with red arrow marks. These distortions can impede the twin growth and slow down the growth process, thus effectively making it a growth-controlled process. Authors should make sure that the growth-controlled twinning is not due to some external barriers to twin growth.

3. Is there any effect of initial nanowire micro-structure on this twin growth transition? This is because the presence of initial defects may lower the nucleation stress (inhomogeneous nucleation) and increase the twin growth stress (by acting as a barrier). Also, the nanowire cross-section shape (circular/square) and side surfaces may also influence the twin growth process. The authors may discuss about the influence of these factors on the twin growth process, if they feel it appropriate and important.

4. Recently, anti-twinning is also reported under the compression of $\langle 110 \rangle$ W nanocrystals of size less than 20 nm (Sci. Adv. 6 (2020) eaay2792). Is there any similar anti-twinning in Ta nanocrystals?

Point by point response to reviewers' comments

(Manuscript ID: NCOMMS-21-34854)

We sincerely thank the reviewers for their insightful comments and constructive suggestions on our manuscript. We have revised our manuscript (highlighted in red font). In the following, we provide a point-by-point response to all the reviewers' comments.

Reviewer #1 (Remarks to the Author):

In the manuscript titled “Atomic-scale” observation of nucleation- and growth-controlled deformation twinning in BCC nanocrystals”, the authors performed in situ atomic-scale TEM and atomistic simulations and reported the mechanism transition of deformation twinning from being “reluctant” or “facile” in BCC nanocrystals. In coarser BCC nanocrystals (>15 nm) deformation twinning is suppressed, whereas twinning is easier in finer BCC crystals (<15 nm). The new knowledge generated from this work focuses on the size-dependent twin nucleation and propagation mechanisms in BCC metals. The experiments are well designed. Results are beautiful. The paper is well written and I recommend acceptance after minor revision.

Response: We thank the reviewer for his/her positive evaluation of our work.

Here are some detailed comments for the authors to consider:

1. In the introduction part, maybe it is a good idea to mention the theoretical strain value deformation twinning in BCC metal could provide. The experimental results acquired later may be compared to the theoretical value. (I see it was mentioned in the results and discussion session. The authors may want to move the information up.)

Response: The theoretical shear strain of deformation twinning in BCC metals is $\sqrt{2}/2 = 0.707$ (Hosford, William F. *The mechanics of crystals and textured polycrystals*. 1993: p. 248). This theoretical shear strain corresponds to the axial tensile strain of 41.4% along [001]. It is very close to the measured 40% elongation in Fig. 2d after the twinned region almost occupied the whole gauge section of the nanocrystal. We have added this information in the Introduction section as the reviewer suggested.

2. In the first scenario (23 nm sample) at the twin embryo stage, is it possible to draw a Burgers circuit to unambiguously identify the Burgers vector of the twin dislocations?

Response: Thanks for the constructive suggestion. It is hard to draw a Burgers circuit to unambiguously identify the Burgers vector of a twinning dislocation due to the small shear displacement ($1/6[111]$) and the limited resolution of TEM images. The Burgers vector marked in Fig. 2b indicates the twinning shear along $[111]$ on the $\{112\}$ plane.

3. It would be great if the authors can include a short paragraph in the manuscript discussing the transition of mechanisms here that may be applicable to twin boundaries in FCC and HCP metals. (The screw dislocation-mediated twin growth vs. facile growth the authors discussed here reminds me of the “shear” vs. “shuffle” debate in Mg twin growth. Maybe something in common may be drawn here?)

Response: To compare the twinning behavior in nanocrystals of different lattice structures, the present BCC Ta nanocrystals exhibits a size-dependent transition from the growth to nucleation-controlled twinning mechanism. This is largely because the continued supply of twinning partials is difficult and the movement of twinning partials on twin boundaries is slow for twin growth in large Ta nanocrystals. As a result, the applied load has to continuously increase to nucleate dislocations, drive cross-slip, and move twinning partials. For FCC crystals, there is no experimental report on the size-dependent transition from the growth to nucleation-controlled twinning mechanism. This indicates that twin growth is less difficult once the applied load has reached the critical value of twin nucleation. For HCP crystals, a different size-dependent twinning behavior was observed (*Yu et al. Nature 463 (2010): 335-338*). As the size of an HCP Ti alloy is reduced to below one micrometer, dislocation slip replaces twinning to dominate plastic deformation. A ‘stimulated slip’ model was used to explain this size dependence of deformation twinning. We have added the above discussion in the revised manuscript (page 14-15).

4. The authors selected Ta, which has a high brittle to ductile transition temperature compared to many other BCC metals. Do authors think other BCC metals but with lower brittle to ductile transition temperatures will exhibit similar observations and also manifest the mechanisms discovered here?

Response: The brittle to ductile transition temperature (BDTT) of Ta is very low ($< 20\text{K}$) (Zhang, M., et al. *Scripta Materialia* 57 (2007): 1032-1035). Hence, the Ta nanocrystals tested are able to generate sufficient plastic deformation without brittle fracture during our in situ straining experiments at room temperature. The BDTTs in other BCC metals are summarized in Table R1. Most BCC metals have the BDTT lower than room temperature, and the sample size is unlikely to cause a drastic change of BDTT. Hence, most BCC metals are expected exhibit a similar size-dependent transition from the growth to nucleation-controlled deformation twinning mechanism, which warrants further systematic studies in the future.

Table R1. The brittle-to ductile transition temperature in BCC metals.

Materials	Transition point (K)	Reference
Fe	~ 140	Tanaka, et al. Acta Materialia 56.18 (2008): 5123-5129.
Mo	~ 230	Hirsch, P. B., et al. Scripta metallurgica et materialia 27 (1992): 1723-1723.
Nb	< 143	Begley, R. T., et al. Journal of the Less Common Metals 3.1 (1961): 1-12.
Cr	343	Lu, Yan, et al. PNAS 118.37 (2021).
W	~ 430	Giannattasio, Armando, and Steve G. Roberts. Philosophical Magazine 87.17 (2007): 2589-2598.
V	~ 160	Giannattasio, A., et al. Phys. Scr 2007.T128 (2007): 87.
Ta	< 20	Zhang, M., et al. Scripta Mater. 57.11 (2007): 1032-1035.

Reviewer #2 (Remarks to the Author):

For several reasons, the Introduction is too general or not specific enough. Mechanical twinning, TWIP effect, and these for different crystal structures (FCC, BCC, HCP) are put on the same level, without necessarily describing the differences. The authors state that the activation of mechanical twinning is in many cases beneficial. It is not clear either what the distinction is between macro- and nanostructures. First of all, in some materials, mechanical twinning is activated to compensate for the lack of gliding systems of dislocations. This is the case for HCP and some BCC. This is quite different from the TWIP effect active in FCC. This mix between classical and nano, twinning and TWIP, BCC and co, etc. does not reflect an adequate perspective of the role of mechanical twinning as a plastic deformation mode.

Response: We greatly appreciate the insightful comments by the reviewer. We agree that the need for twinning is different in HCP, BCC and FCC crystals. This work is focused on the size effect on the transition from growth to nucleation-controlled deformation twinning in BCC nanocrystals. We have revised the paper to distinguish the roles of twinning in different crystal structures (e.g., see the new paragraph added on pages 14-15). To avoid confusion with the TWIP effect on bulk materials, we provide explicit discussions on twinning and dislocation plasticity rather than using the term TWIP in the revised Discussion section.

At line 58, the authors state “This result indicates that the limited TWIP effect in BCC metals may not only be attributed to the high critical stress required for twin nucleation, but also to the slow process of twin growth.” This sentence is barely comprehensible. What do the authors mean by 'growth'? Is it lengthening or thickening? Considering for example the TWIP effect in some titanium alloys like Ti-15%Mo, this statement is difficult to consider.

Response: We have revised this sentence as follows. “This result indicates that the limited twinning deformation in BCC metals may be attributed to not only the high critical stress required for twin nucleation, but also the even higher stress needed for twin growth (i.e., thickening and lateral expansion of a twin embryo).”

The results presented are interesting, but I have some doubts about the consistency of the complete story and the accuracy of the measurements made in relation to the conclusions drawn. The procedure used for the fabrication of the samples used for the in-situ traction is quite particular, in

relation to the nature of the material studied (Ta). But the consequence is a geometry of the tensile specimens that does not correspond in all likelihood to a uniaxial tensile test and presents geometrical defects that undoubtedly influence the conditions of nucleation and growth of the twins. From what it can be seen on the different pictures illustrating the traction, the specimens are strongly constrained and it is quite sure that the stress state is not the same when the thickness changes. For example, Fig.1 and associated movie strongly looks like a localization mechanism due to a stress state resulting from some confinement. It is indeed quite evident that the width has an influence once considering the coordinated movement involved in growth as shown on Fig.2. As a consequence, the concluded crystal size effect could be a geometry effect.

Response: Thanks for the insightful comments. As the reviewer noted, the sample geometry can affect the stress state in nanocrystals. To assess the impact of sample geometry and surface defects on the observed twinning modes, we performed in situ straining experiments on nanocrystals with similar diameters but different initial geometries. As shown in Fig. 2, the two Ta nanocrystals had sub-15 nm diameters yet different initial shapes, both exhibiting nucleation-controlled twinning under $\langle 001 \rangle$ tension. Moreover, we conducted similar staining experiments on W nanocrystals. As shown in Fig. R1 of this Response, two W crystals with diameters of 8.3 and 6.8 nm exhibited a consistent nucleation-controlled twinning mode under compression along $[110]$ (equivalent to $\langle 001 \rangle$ tension due to Poisson's effect). The two W crystals had different geometries and surface step characteristics (including variations in step sizes and distributions) but revealed the same twinning mode. These experimental results collectively indicate that the size effect plays a predominant role in determining the twinning mode of BCC nanocrystals, as compared to the influence of sample geometry and surface defects.

Fig. R1 (a) Compression-induced twinning in a ~8.3 nm W nanocrystal. (b) Compression-induced twinning in a ~6.8 nm W nanocrystal.

Still considering the reported cases of twin nucleation, an accurate characterization of the configuration of the defect(s) bringing the embryo would be needed to understand the different reported situations. It would be worth elucidating if these defects result from the ‘machining’ of the specimens or if they are intrinsic defects.

Response: Thanks for the thoughtful comments. Based on prior research (*Wang et al. Nature materials 14 (2015): 594-600*), it is likely that the surface formation of twin embryos is controlled by the surface nucleation of dislocations. However, it is hard to detect and track the surface nucleation of dislocations in BCC nanocrystals through high-resolution TEM imaging. As

discussed above, the initial sample geometry and surface defects (e.g., different step sizes and distributions) did not exert significant influence on the twinning modes in both Ta and W nanocrystals. In fact, the sample size predominantly governs the transition from a growth-controlled to a nucleation-controlled twinning mechanism.

Line 118, Looking at Fig S5 and Fig1, they are not presenting the same thing, contrarily to what is suggested.

Response: In Fig. S5b, we compare the elongation over time during deformation twinning in the 23-nm-diameter Ta nanocrystal in Fig. 1 (see the magenta curve with the tensile strain given on the right y axis) and the 10.4-nm-diameter Ta nanocrystal in Fig. 2a-d (see the dark cyan curve with the tensile strain given on the left y axis). For the 23-nm-diameter Ta nanocrystal, its plastic deformation was primarily governed by growth-controlled twinning (Fig. 1), and no strain burst was observed. For the 10.4-nm-diameter Ta nanocrystal, its plastic deformation was dominated by nucleation-controlled twinning (Fig. 2), and a strain burst accompanying twin nucleation was observed at ~9%.

In the beginning of the Discussion, it is stated that mechanical twinning ‘replaces’ dislocation glide. This is thus not a TWIP effect, which involves a very large dislocation activity.

Response: Our *in situ* observations indicate that mechanical twinning predominantly controlled the plastic deformation of Ta nanocrystals in our experiments. Nevertheless, abundant dislocation activities were also captured during the twinning process in large Ta nanocrystals with diameters greater than 15 nm (Fig. 1). This is because the increased stress during the reluctant growth of twins can trigger the activation of dislocation plasticity. To avoid any potential confusion with the TWIP effect on bulk materials, we have provided explicit discussions on twinning and dislocation plasticity rather than using the term TWIP in the revised manuscript.

Reviewer #3 (Remarks to the Author):

Review report

Journal : Nature Communications

Title: Atomic-scale observation of nucleation- and growth-controlled deformation twinning in body-centered cubic nanocrystals

In BCC nanowires with size less than 50 nm, twinning is an important mode of plastic deformation. However, still there is a lack of understanding on the details of twin nucleation and growth, and size effects on twinning related phenomena, especially from the experimental perspective. In this regard, the present paper makes a significant improvement towards our understanding of twinning in BCC nanowires. Using in-situ TEM and atomistic simulations, the authors show a transition in twin growth mechanism with respect to nanocrystal size. They show that, in relatively smaller size nanocrystals (< 15 nm), the twinning is nucleation controlled and hence its growth is facile, whereas in nanocrystal with size > 15 nm, the twinning is growth controlled (reluctant twin growth). Apart from this, the authors also show a dislocation controlled plasticity in nanocrystals having an antitwinning sense on the most favoured $\{112\}$ planes. The authors clearly explained and discussed all the results in detail and provided the reasons for the observed transition from facile to reluctant twin growth with increasing size. Overall, an excellent paper and it can be accepted after incorporating the following minor suggestions/questions;

Response: We thank the reviewer for his/her positive evaluation of our work.

1. I feel that it would be nice, if the authors show the transition quantitatively. The authors have shown that in nucleation-controlled twinning, twin growth stress is much lower than the twin nucleation stress (Figure 2h). Similarly, if possible, it should be shown that, in growth controlled twinning, the twin migration stress or stress during twin growth should be greater than the twin nucleation stress. If not experimentally, it is definitely possible through atomistic simulations. Previous study focused on twinning and twin growth in BCC Fe nanowires (Comput. Mater. Sci. 104 (2015) 76-83) has shown that up to 24.27 nm size, the Fe nanowires show nucleation controlled twinning. As a result the migration stress is always lower than the twin nucleation stress up to the size of 24.27 nm. Further, limited ductility has been reported in relatively thicker nanowires due to difficulty in twin growth along the nanowire length.

Response: For the sub-15-nm-diameter Ta nanocrystal in Fig. 2h, the twin nucleation stress is ~ 6.8 GPa, much higher than the twin growth stress ~ 2.8 GPa. For the 23nm-diameter Ta nanocrystal in Fig. 1, the twin nucleation stress is ~ 4.6 GPa and the applied load needs to increase for continued twin growth. Due to the timescale limitations of molecular dynamics (MD) simulations,

the growth-controlled twinning processes could not be obtained by direct MD simulations. The MD paper noted by the reviewer is helpful for understanding the nucleation-controlled twinning process and we have added the citation of this paper in the revised manuscript.

2. In Figure 1 (c-f), the authors have highlighted some distortions in the vicinity of twin with red arrow marks. This distortions can impede the twin growth and slow down the growth process, thus effectively making it growth controlled process. Authors should make sure that the growth controlled twinning is not due to some external barriers to twin growth.

Response: The lattice contrast or the equal inclination fringe on the TEM images was not formed before twinning, and it was induced by lattice bending and distortion in the vicinity of growing twins. Due to the twinning-induced crystal reorientation and the existence of inclined twin boundaries, the resulting distortions near the twin can impose high barriers to twin growth compared to some external barriers such as pre-existing dislocations.

3. Is there any effect of initial nanowire microstructure on this twin growth transition? This is because, the presense of an initial defects may lower the nucleation stress (inhomogenous nucleation) and increase the twin growth stress (by acting as a barriers). Also, the nanowire cross-section shape (circular/square) and side surfaces may also influence the twin growth process. The authors may discuss about the influence of these factors on twin growth process, if they feel it appropriate and important.

Response: Thanks for the constructive comments. To evaluate the effect of sample geometry and surface defects on the observed twinning modes, we performed in situ straining experiments in nanocrystals with similar diameters but different initial geometries. As shown in Fig. 2, the two Ta nanocrystals had sub-15 nm diameters but different initial shapes, and they showed the nucleation-controlled twinning under $\langle 001 \rangle$ tension. Moreover, we conducted the similar staining experiments of W nanocrystals. As shown in Fig. R1, two W crystals with 8.3 and 6.8 nm diameters exhibited a similar nucleation-controlled twinning mode under compression along $[110]$ (equivalent to $\langle 001 \rangle$ tension due to Poisson's effect). The two W crystals had different geometries and surface step characteristics (e.g., different step sizes and distributions) but showed the same twinning mode. These experimental results indicate that the size effect plays a more important role in the twinning mode of BCC nanocrystals compared to the sample geometry and surface defects.

4. Recently, an anti-twinning is also reported under the compression of W nanocrystals of size less than 20 nm (Sci. Adv. 6 (2020) eaay2792). Is there any similar anti-twinning in Ta nanocrystals?

Response: Anti-twinning is not observed during our in situ straining experiments for Ta nanocrystals. In the Science Advances paper, anti-twinning is found during $\langle 110 \rangle$ tension (equivalent to $\langle 001 \rangle$ compression due to Poisson's effect). For $\langle 001 \rangle$ tension in our experiments, normal twinning occurs along the $\langle 110 \rangle$ shear direction on the $\{112\}$ twin plane (Figs. 1 and 2).

REVIEWER COMMENTS

Reviewer #1 (Remarks to the Author):

The authors have adequately addressed all my comments.

Upon the editor's request, I also went through the comments from Reviewer #2 and the responses from the authors.

One concern is that the twin nucleation behavior is not governed by the crystal size, rather it is influenced by the surface geometric irregularity or defects at the atomic scale. In the response, the authors demonstrated that even with different initial geometries. The authors also demonstrated a similar observation made in W.

One thing the authors did not address though was that they looked at the width but not the thickness. The authors should tilt the sample at a few large angles (like tomography but no need to do a complete one) to demonstrate that smaller-width samples are also thinner in thickness or show that the samples are indeed circular in cross-section so one parameter can describe it.

I consider the comment regarding surface defects raised by Reviewer #2 to be an astute one. Here are just some thoughts from my end. The twin nucleation is heterogeneous and takes place from a defect on the crystal surface. Assuming the samples are prepared in the same way, the surface defect characteristics should be comparable. Then twin nucleation behavior should be governed by the surface defect statistics, the observed differences can be attributed to the size effect. (Size effect is a result of defect statistics in a given volume or surface area.)

Reviewer #2 (Remarks to the Author):

I carefully read the answers to the comments raised at the first submission, together with the modifications brought to the manuscript. While some strong comments were raised, I think, minor modifications were brought. Particularly, I do not see how the 2 TEM micrographs given in the responses to reviewers demonstrate against the comment raised about the influence of the geometry of the specimens.

On the other hand, I do not agree with the logic of the added sentence "For FCC crystals, there is no experimental report on the size-dependent transition from the growth to nucleation-controlled twinning mechanism. This indicates that twin growth is less difficult once the applied load has reached the critical value of twin nucleation." This is not because a specific phenomenon has not been reported (or even studied) up to now that the opposite is true.

Reviewer #3 (Remarks to the Author):

The have answered satisfactorily to the comments raised by this reviewer. Hence, the manuscript may be considered for publishing.

Point-by-point response to reviewers' comments

(Manuscript ID: NCOMMS-21-34854A)

We sincerely thank the reviewers for their insightful comments and constructive suggestions. Revisions have been made to further improve the quality of our manuscript. For clarity, changes made to the revised text are highlighted in yellow. In the following, we provide a point-by-point response to the reviewers' comments.

Reviewer #1 (Remarks to the Author):

The authors have adequately addressed all my comments.

Upon the editor's request, I also went through the comments from Reviewer #2 and the responses from the authors.

Response: We appreciate the reviewer's careful evaluation of our revised manuscript.

One concern is that the twin nucleation behavior is not governed by the crystal size, rather it is influenced by the surface geometric irregularity or defects at the atomic scale. In the response, the authors demonstrated that even with different initial geometries. The authors also demonstrated a similar observation made in W.

One thing the authors did not address though was that they looked at the width but not the thickness. The authors should tilt the sample at a few large angles (like tomography but no need to do a complete one) to demonstrate that smaller-width samples are also thinner in thickness or show that the samples are indeed circular in cross-section so one parameter can describe it.

Response: We thank the reviewer for his/her constructive suggestions. We have performed additional experiments to investigate the cross-sectional geometry of the samples. As shown in Fig. R1, only ~10% difference in diameter is measured when the sample is gradually tilted across a span of 50°, corroborating a near-circular shaped cross section. The circular cross section of the samples in the present work can be attributed to the applied ultrafast liquid-quenching sample preparation technique (Nature 512, 177-180, 2014). Since the samples are solidified from an intermediate liquid state, a circular cross section tends to form to minimize surface energy. Figure

R1 has been added to the revised manuscript as Supplementary Fig. S9, and the text is revised accordingly (highlighted in yellow on page 17).

Figure R1. Near-circular cross-section of a typical specimen prepared by the ultrafast liquid-quenching approach. Only ~10% difference in diameter is measured when the sample is gradually tilted across a span of 50°. The scale bars are 10 nm.

I consider the comment regarding surface defects raised by Reviewer #2 to be an astute one. Here are just some thoughts from my end. The twin nucleation is heterogeneous and takes place from a defect on the crystal surface. Assuming the samples are prepared in the same way, the surface defect characteristics should be comparable. Then twin nucleation behavior should be governed by the surface defect statistics, the observed differences can be attributed to the size effect. (Size effect is a result of defect statistics in a given volume or surface area.)

Response: We appreciate the reviewer's insightful comments on the observed size effect of the twinning mode. As shown in Fig. R2, samples prepared in the present study share comparable geometries and defect characteristics. Specifically, surface defects consist almost exclusively of atomic steps with variations in step height and distribution (Fig. R2c, d). As a result, the size dependence of the surface defect statistics should result in the size-dependent deformation behavior. We have now included Fig. R2 in the revised manuscript as Supplementary Fig. S10, and revised the text accordingly (highlighted in yellow on page 17).

Figure R2. Comparable geometry and surface defect characteristics between specimens in the current study. a-b, Two as-formed specimens with similar geometries. The scale bars are 2 nm. c-d, Enlarged views of red-boxed regions in (a) and (b). The surface defects of both samples are almost exclusively atomic steps, with a slight difference in distribution.

Reviewer #2 (Remarks to the Author):

I carefully read the answers to the comments raised at the first submission, together with the modifications brought to the manuscript. While some strong comments were raised, I think, minor modifications were brought. Particularly, I do not see how the 2 TEM micrographs given in the responses to reviewers demonstrate against the comment raised about the influence of the geometry of the specimens.

Response: We thank the reviewer for the critical review of our work. Indeed, the mechanical testing conditions in the present work are not identical between each experiment, owing to the technical difficulties associated with nanomechanical testing. For example, differences may be present in specimen shape, aspect ratio, loading direction, surface defect configuration, and so forth. Hence, we agree with the reviewer that factors other than specimen size may play a role in determining the deformation behavior, and thus should not be overlooked. To address this issue, we have performed a number of experiments in order to evaluate the size effect. On one hand, we

have demonstrated in the previous response letter that slight differences in the sample geometry and surface defect configuration are not sufficient to change the primary deformation mode, which is the same for both Ta and W samples. This could be understood by taking into account the comparable geometries and defect characteristics of specimens (see Fig. R2), which connect the size effect of defect statistics with that of deformation behavior. On the other hand, we have also evaluated the statistics regarding the relationship between specimen diameter and deformation mode (i.e., twinning vs. dislocation plasticity), which clearly reveal a decreasing tendency for twin nucleation with increasing specimen diameter (Fig. R3). This new result confirms the predominant impact of the size effect on deformation mode.

Figure R3. A number of experiments reveal the size dependence of deformation mode, i.e., twinning vs. dislocation plasticity. A decreasing tendency for twin nucleation is observed with increasing specimen diameter.

On the other hand, I do not agree with the logic of the added sentence "For FCC crystals, there is no experimental report on the size-dependent transition from the growth to nucleation-controlled twinning mechanism. This indicates that twin growth is less difficult once the applied load has reached the critical value of twin nucleation." This is not because a specific phenomenon has not been reported (or even studied) up to now that the opposite is true.

Response: We appreciate the reviewer’s careful evaluation of our manuscript. We have revised the sentences as follows: “For FCC crystals, such a size-dependent transition from growth- to nucleation-controlled deformation twinning is absent in various experimental studies (Nature Communications 1, 144, 2010; Nature Communications 5, 3033, 2014; Nano Letters 11, 3499-3502, 2011), indicating that twin growth is less difficult once the applied load has reached the critical value for twin nucleation.” (highlighted in yellow on page 15)

REVIEWERS' COMMENTS

Reviewer #1 (Remarks to the Author):

The authors have successfully addressed all my comments and I recommend accepting the paper in the current form.

Reviewer #2 (Remarks to the Author):

The authors adequately answered to my raised comments. The complementary experiments are greatly appreciated.